# Is the Beck Depression Inventory (BDI) a Useful Tool for Predicting IVF Success?

**DOI:** 10.3390/medicina61010099

**Published:** 2025-01-10

**Authors:** Dragiša Šljivancanin, Snežana Vidaković, Darija Kisić Tepavčević, Bojana Petrović, Una Šljivančanin, Milan Dokić

**Affiliations:** 1Clinic for Obstetrics and Gynecology, 11000 Belgrade, Serbia; drvidakovicsnezana@gmail.com (S.V.); mrdrbojaninmail@gmail.com (B.P.); milanddokic@gmail.com (M.D.); 2Obstetrics and Gynecology Department, Faculty of Medicine, University of Belgrade, 11000 Belgrade, Serbia; darijakt@gmail.com; 3Institute of Epidemiology, 11000 Belgrade, Serbia

**Keywords:** in vitro fertilization, depression, Beck’s depression inventory

## Abstract

*Background and Objectives*: Infertility is a global problem. The interaction between depression and infertility seems bidirectional, and depression may negatively influence IVF outcomes. The Beck Depression Inventory (BDI) is one of the most extensively used instruments for diagnosing depression. The aim of this study was to assess the dynamics of depression in patients undergoing IVF utilizing the BDI. *Materials and Methods*: This prospective cohort study was conducted at the Clinic for Gynecology and Obstetrics of the University Clinical Center of Serbia in Belgrade, Serbia during the year 2019. Both partners in the IVF program were required to complete the paper-based BDI 5 to 10 days before starting ovarian stimulation, within the 48 h preceding oocyte aspiration and four weeks after embryo transfer. We sought to identify potential predictors of a favorable IVF outcome by using logistic regression modeling. *Results*: Our study enrolled 86 couples. The overall IVF efficacy in our cohort was 18.6%. A statistically significant logistic regression model (*p* = 0.001) managed to explain 47.6% of the variability. Increasing patient age, the presence of depression (BDI > 16) and the number of failed IVF cycles were found to be significant negative predictors of ongoing IVF success. *Conclusions*: Depression is more common in patients undergoing IVF than in the general population. Depression might increase the odds of IVF failure, lending credence to the idea that depression screening using BDI should be a routine part of the IVF process.

## 1. Introduction

Infertility is a global problem with some regions of the world reporting as much as a quarter of its reproductive age population being affected [1]. It is estimated that infertility treatment costs in France totaled approximately EUR 300 million in 2022 alone [2]. The precise prevalence of infertility in Serbia is not known. However, the total fertility rate in Serbia was estimated at 1.6 in 2022, which is almost identical to the European average of 1.53 [3,4].

Since infertility is primarily a consequence of organic pathology, most (patients and physicians alike) tend to downplay the psychologic stresses that these patients endure on a daily basis. The most common among these are fear of treatment failure, fear of the adverse events related to infertility treatments, a feeling of hopelessness concerning the ability to establish a family, etc. [5]. If such stresses persist and compensatory mechanisms are not strong enough to overcome them, depression may ensue. The longer the number of years living with infertility and the longer the period of infertility treatment, the higher the proportion of patients with depression [6].

In vitro fertilization (IVF), for some couples, comes relatively late in the infertility treatment process, at a delicate moment when they are already under psychologic stress at IVF inception [7]. A group of researchers has found that a quarter of women and a tenth of the men starting IVF had already experienced a depressive episode. The situation was found to have worsened with the process itself with half the women and a third of the men experiencing at least a single depressive episode by the end of IVF [8]. Patients have been found to consider IVF more psychologically than physically challenging, with half the patients who have underwent IVF considering it to have been the most stressful experience of their lives [9]. Couples who ultimately fail to become parents are at increased risk of schizophrenia, other psychoses, substance abuse, anxiety, depression and eating disorders [10]. Perhaps surprisingly, even couples who had undergone successful IVF were found to be more depressed than those that had conceived naturally [11]. Finally, we have come to understand that the interaction between depression and infertility is a two-way one, and that depression can negatively influence IVF outcomes [12,13].

In Europe, depressive disorders are currently diagnosed using the International Classification of Diseases 11th Revision (ICD-11) definitions which are almost identical to the definitions employed in the US and outlined in the Diagnostic and Statistical Manual of Mental Disorders, Fifth Edition (DSM-5) [14]. Numerous instruments for screening for depression have been devised, with the Beck Depression Inventory (BDI) being one of the most extensively used [15]. The instrument comprises 21 self-reported items devised as a four-point Likert scale. Thus, a patient can have a BDI score from 0 to 63, with most experts considering a score higher than 16 a sign of clinical depression [16].

The aim of this study was to assess the dynamics of depression in patients undergoing IVF by utilizing the BDI at three distinct points in time—just before ovarian stimulation, just before oocyte aspiration and four weeks after embryo transfer is completed. The secondary goal of this study was to determine the impact depression severity before and during IVF has on its success.

## 2. Materials and Methods

This prospective cohort study was conducted at the Clinic for Gynecology and Obstetrics of the University Clinical Center of Serbia in Belgrade, Serbia. We enrolled all couples starting state funded IVF at our clinic during the year 2019. All patients (both partners) enrolled in the study were required to complete the paper-based BDI 5 to 10 days before starting ovarian stimulation, within the 48 h preceding oocyte aspiration and four weeks after embryo transfer. The BDI was calculated by applying the methodology that has previously been published by Beck et al. [16]. Depression was defined as a score of more than 16. Couples with a history of psychiatric and malignant diseases were excluded from the study.

Demographic data (gender, age, education level, employment status), data on the causes of infertility and previous infertility treatments (period living with infertility, previous IVF attempts) were collected from all patients. Additionally, data on the characteristics of the current IVF cycle were also compiled—IVF protocol type and IVF outcome.

We used descriptive statistics to analyze demographic characteristic as well as infertility-related medical histories and current IVF methods. Means and standard deviations were used to describe continuous normally distributed variables, while medians and ranges to describe continuous variables that are not normally distributed. Categorical data were analyzed using ratios and percentages.

When comparing more than two dependent normally distributed continuous variables, a repeated measures ANOVA was used. For more than two dependent non-normally distributed continuous variables, a Friedman test was employed. Independent categorical variables were compared using a chi-square test, while a McNemar test was used if the variables were related. When the assumptions of the chi-square test were violated, a Fisher test was utilized.

We sought to identify potential predictors of a favorable IVF outcome by using multivariable logistic regression modeling. Our model included the following predictors: age, employment status, education level, years of infertility, number of failed IVF cycles, current number of transferred embryos and depression before current IVF onset.

We used IBM’s SPSS Statistics v23 (IBM, Armonk, New York, USA) to conduct our statistical analyses. Statistical significance was set at 0.05.

All study participants signed an informed consent form. The study was approved by the Ethical Committee of the University Clinical Center of Serbia (29/II-7, 25 February 2016).

## 3. Results

Our study enrolled 86 couples (172 patients in all). Their demographic data and personal medical histories are graphically represented in Table 1. The mean age in our cohort was found to be 34.7 ± 4 years and 13.9% of patients were highly educated. Men and women did not differ significantly when it came to age (*p* = 0.64) and education levels (*p* = 0.32). A total of 58 men (67.4%) and 34 women (39.5%) were 35 years old or older at study enrollment. For 28 couples (32.6%), this was their first IVF attempt.

The causes of infertility in our cohort are represented in Figure 1. The most common cause in our cohort was isolated male infertility, which was detected in 36 (41.8%) couples, while both male and female infertility was detected in 14 couples (16.3%). A total of 30 men had a low sperm count, while azoospermia was detected in 24 men. Anovulation due to causes other than PCOS was demonstrated in five women and was the only cause of infertility in all of these couples. In two couples, a low ovarian reserve was found to be the only cause of infertility. Two women, classified to other causes of infertility, had cervical causes of infertility.

In our institution, we usually utilize a short personalized GnRH antagonist protocol wherein cetrorelix-acetate is used for LH suppression. Ovarian stimulation is achieved using follitropin alfa. Ovulation is triggered using choriogonadotropin alfa. All patients in this study received the aforementioned drugs. After IVF was completed, 24 pregnancies were confirmed and 16 children were born. Thus, the overall IVF efficacy in our cohort was 18.6%.

BDI was higher than 16 (the cut-off for depression) in 17.4%, 20.3% and 43.6% before, during and after IVF, respectively. We found a statistically significant difference between the prevalence of depression before starting IVF and four weeks after embryo transfer (McNemar test, *p* = 0.002). The mean BDI scores before IVF, during IVF and after IVF were 12.1 ± 5.3, 13.6 ± 6 and 15.3 ± 7, respectively. We found this difference to be statistically significant using the repeated measures ANOVA test (F = 11.437, *p* = 0.032). A post hoc Bonferroni analysis revealed significant differences between BDI scores before IVF and during IVF (*p* = 0.041), during and after IVF (*p* = 0.031), as well as before and after IVF (*p* = 0.038).

As is evident from Figure 2, depression levels rose non-significantly in men during the course of IVF. In females a significant difference between BDI scores in the days before ovarian stimulation and weeks after embryo transfer (*p* = 0.04) as well as between the BDI scores before IVF and during IVF (*p* = 0.01) was observed.

The existence of depression before IVF onset has been associated with an unfavorable IVF outcome in our study (χ^2^ = 7.93, *p* = 0.04). The same holds true for patients that were depressed during IVF (χ^2^ = 6.89, *p* = 0.06). However, BDI scores at both IVF inception and during the cycle itself did not differ significantly between those that failed IVF and those that did not—*p* = 0.22 and *p* = 0.29, respectively.

We utilized multivariable logistic regression modeling that included the variables enumerated in Table 2 to try to explain the variability of IVF success in our study. The model was statistically significant and managed to explain 47.6% of the variability (Nagelkerke R^2^ = 0.476, *p* = 0.001). Increasing patient age, the presence of depression (BDI > 16) and the number of failed previous IVF cycles were found to be the only statistically significant negative predictors of current IVF success.

When depression was substituted for the BDI score as a predictor of IVF failure, the logistic regression model remained statistically significant (*p* = 0.032), but the model was able to accurately predict only 38.3% of the variability (Nagelkerke R^2^ = 0.383). The BDI score was not a significant predictor of IVF failure (*p* = 0.11). Increasing age and increasing number of failed IVF attempts remained significant predictors of IVF failure.

## 4. Discussion

The average age in our study was 34.7 years, with women being somewhat younger (33.6 vs. 35.8 years). This is not surprising given that, worldwide, men are typically older than women at the moment their first child outside the IVF context [17]. The ages of the female patients in our study were similar to those reported by other authors [18]. A high percentage (39.5%) were 35 or older.

The level of education was higher in women—16.3% women had a college degree, while the same level of education was achieved by 11.6% of the men in our study. A group of Chinese researchers reported that a little less than half of the female patients in their IVF cohort had a college degree [19]. In the general population of Europe, 36% of women and 31% of men have a tertiary degree [20]. The unemployment rate in our cohort was in line with the unemployment rate in the general population of Serbia [21].

The couples in our cohort had been treated for infertility for 1 to 7 years (median 3 years) before study enrollment. Other scientists have reported similar data [22]. Approximately a third (32.6%) of our patients never underwent IVF before study enrollment. The median number of previous IVF cycles was 1 in our cohort. The largest number of failed cycles was 5, which was observed in 2 couples. Similar observations were published by others [23].

The different causes of infertility in our cohort were distributed in a way that is, for the most part, very similar to those reported by other researchers [24]. The most common isolated cause of infertility in our study is male infertility, and this is commonly reported by others [24]. Tubal factor infertility was the second most commonly reported factor. Various studies have reported radically different prevalences of this type of infertility ranging from 10% to 60% [24]. We detected only a small number of patients with infertility due to PCOS, ovarian dysfunction and endometriosis, which was also observed in other studies [24]. Infertility of an unknown cause has previously been reported in 5 to 25% of patients, and our study is no exception [25].

Sixteen couples (18.6%) succeeded in becoming parents after the IVF cycle analyzed in this study. This almost equivalent to the efficacy reported in Europe in 2018, which was estimated to be 19.6% [26].

Depression was present (using a BDI score of 16 or higher) in 17.4% of patients before IVF, in 20.3% of patients during IVF and in 43.6% after IVF. This difference proved statistically significant when the period before and after IVF were compared (*p* = 0.002). Thus, depression was at least five times more prevalent in our study than in the general population [27]. The median BDI score five days before ovarian stimulation was 12.1, 13.6 just before oocyst aspiration and 15.3 four weeks after embryo transfer. These differences were found to be statistically significant (*p* = 0.032). A post hoc analysis revealed that the differences remained significant when BDI scores before and during IVF (*p* = 0.041), before and after IVF (*p* = 0.038) and during and after IVF were compared (*p* = 0.031). When these differences were analyzed by gender only, the women demonstrated a significantly higher BDI score when scores before IVF and during IVF (*p* = 0.01), as well as the scores before and after IVF (*p* = 0.04), were compared. Even though most other studies on the topic demonstrated a lower BDI than was observed in our patients, some studies found BDI scores similar to and even higher than we did [12]. Other have also demonstrated women have higher BDI scores in the IVF context [28]. Emery et al. have also demonstrated an increase in BDI scores between the time before starting IVF and 6 weeks after embryo transfer [28].

Our logistic regression model aimed at identifying predictors of favorable IVF outcomes was statistically significant (*p* = 0.001) but was able to explain only 57.6% of the variability in the system. Advancing age (*p* = 0.01), an increasing number of previous unsuccessful IVF cycles (*p* = 0.02) and depression diagnosed by an elevated BDI proved to be significant predictors of IVF failure in our study. A meta-analysis by van Loendersloot et al. also found that increasing age is a predictor of IVF failure. They also found that an increasing duration of infertility is another risk factor for IVF failure, along with an elevated baseline level of FSH, and a lower count and/or quality of oocytes [29]. However, others did not find any predictors of IVF failure in their logistic regression model [30]. No studies that we are aware of demonstrated that depression is a predictor of IVF failure. We found two studies in which composite variables derived by combining various emotional stresses like depression and anxiety were evaluated as predictors of IVF success. These two studies yielded radically different results [12,13].

All this being said, it might be prudent to evaluate the value of psychological support for couples undergoing IVF in future research—not just because it might exerts a positive influence on IVF outcomes but also because it might positively affects a couple’s quality of life.

Our study has several limitations. First, the sample size was small, limiting the generalizability of our observations. Hence, our results should be viewed primarily as a proof-of concept. Further studies, on larger samples, would provide more statistical power and as such might offer greater certainty to our conclusions. In addition, we recognize that correlation does not equate with causation. Thus, a correlation between IVF outcomes and depression should be explored further in order to exclude hereto-unrecognized confounding factors like hormone levels, lifestyle factors (diet, exercise, smoking, etc.). Our study was also biased since male infertility was the most common etiology of infertility, which is a consequence of a relatively small sample size.

## 5. Conclusions

IVF is stressful for both partners. It is not surprising then that depression was more common in patients undergoing IVF than in the general population. Worrisomely, depression might increase the odds of IVF failure, lending credence to the idea that depression screening using BDI should be a routine part of the IVF process.

## Figures and Tables

**Figure 1 medicina-61-00099-f001:**
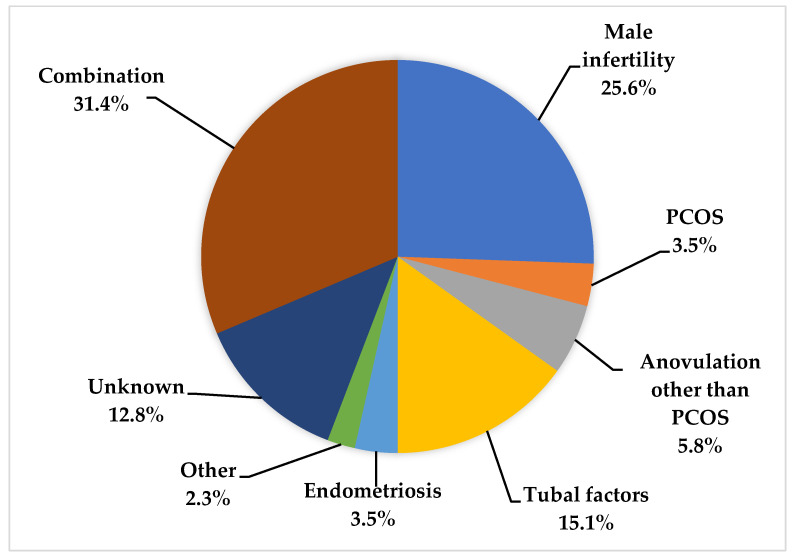
Causes of infertility.

**Figure 2 medicina-61-00099-f002:**
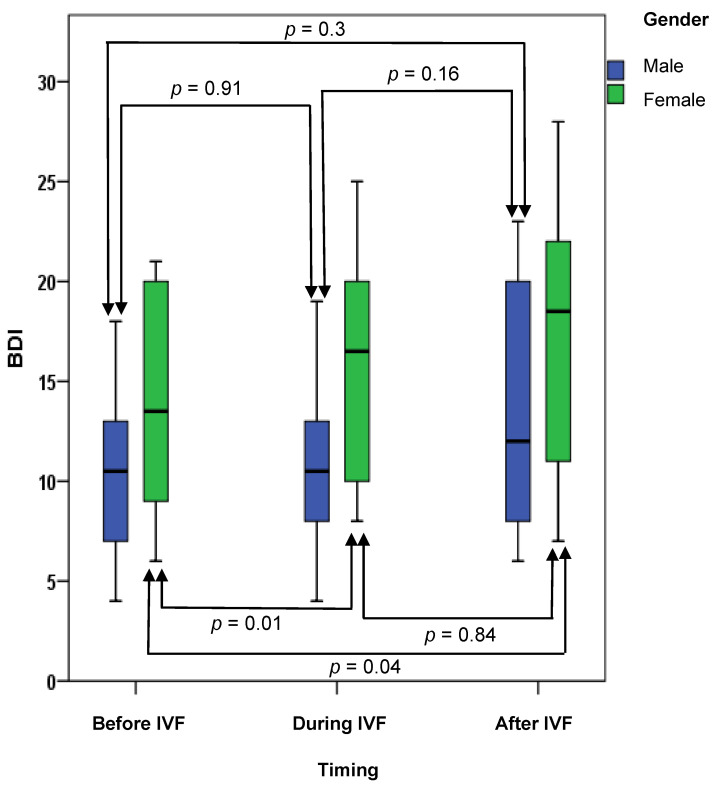
BDI scores by gender in different moments related to IVF.

**Table 1 medicina-61-00099-t001:** Patient demography and medical history.

	Male	Female
Characteristics		
$\mathrm{Age},\;\bar{x}\;$± SD	35.8 ± 4.1	33.6 ± 3.8
Higher education, % (ratio)	11.6% (10/86)	16.3% (14/86)
Both partners employed, % (ratio)	83.7% (72/86)	
Years of infertility, median (range)	3 (1–7)	
Previous IVF cycles, median (range)	1 (0–5)	

**Table 2 medicina-61-00099-t002:** Multivariable logistic regression model for predicting IVF success.

Characteristic	OR (CI 95%)	*p*
Age	0.4 (0.2–0.82)	0.01
Unemployment	1.1 (0.91–2.84)	0.54
Higher education	1.93 (0.72–5.45)	0.3
Years with infertility	1.1 (0.98–1.12)	0.12
Previous IVF cycles	0.67 (0.30–0.79)	0.02
Number of transferred embryos	1.11 (0.99–1.14)	0.15
BDI > 16 before IVF	0.81 (0.72–0.96)	0.04

## Data Availability

The data presented in this study are available on request from the corresponding author due to privacy and ethical restrictions.

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
