# Peer review of "Is the Beck Depression Inventory (BDI) a Useful Tool for Predicting IVF Success?"

_medicina, 2025, doi:10.3390/medicina61010099_

Round 1

Reviewer 1 Report

Comments and Suggestions for Authors

The article you uploaded focuses on the relationship between depression and IVF success, particularly using the Beck Depression Inventory (BDI) as a tool for screening depression in couples undergoing IVF treatment. The originality and relevance of the paper to the field, especially concerning the gaps it addresses, can be summarized as follows:

The paper investigates the dynamics of depression during different stages of the IVF process, which is not widely explored. It assesses depression at three key points before ovarian stimulation, before oocyte aspiration, and after embryo transfer using BDI scores. This approach helps in understanding how depression fluctuates throughout the IVF process.

In summary, the study’s originality lies in its focus on the timing of depression screening during IVF treatment and its implication that addressing mental health could improve IVF outcomes. It fills a gap in current IVF practices, which tend to focus on physical factors and overlook the psychological aspects that could influence success rates.

From the article, several methodological improvements and further controls could be considered for future studies. Here are the recommendations:

1.     1.The authors mention that the sample size of their study is small, which limits the generalizability of their results. Increasing the sample size in future studies would improve the reliability of the findings and provide more robust statistical power for detecting significant predictors of IVF outcomes.

2.     The study recognizes that correlation does not equate to causation. Further controls should be considered to account for potential confounding variables not included in the study, such as the severity of infertility, hormonal levels, lifestyle factors (diet, exercise, smoking), and psychological interventions that might influence depression levels and IVF outcomes.

By addressing these limitations, the study could yield stronger evidence and more comprehensive insights into the relationship between depression and IVF outcomes.

Author Response

Comment 1: The authors mention that the sample size of their study is small, which limits the generalizability of their results. Increasing the sample size in future studies would improve the reliability of the findings and provide more robust statistical power for detecting significant predictors of IVF outcomes.

Answer 1: We agree and we have made the necessary modifications to the manuscrtipt (page 6, paragraph 6, lines 227 through 230.

Comment 2: The study recognizes that correlation does not equate to causation. Further controls should be considered to account for potential confounding variables not included in the study, such as the severity of infertility, hormonal levels, lifestyle factors (diet, exercise, smoking), and psychological interventions that might influence depression levels and IVF outcomes.

Answer 2: We agree and we have made the necessary modifications to the manuscrtipt (page 6, paragraph 6, lines 231 through 233.

Reviewer 2 Report

Comments and Suggestions for Authors

line 42: The in vitro fertilization is not the final step of infertility treatment it is the first step in many cases.

I would compare the changes in depression among people who feel guilty for the cause of infertility - men with a male factor with men without a male factor. And healthy women with women with a strong infertility factor.

Also, the difference in years of effort should significantly affect the rate of depression. Therefore, the study group is heterogeneous and should be divided and studied separately.

The conclusion that  "one must consider psychological support paramount for couples undergoing IVF, not just because it exerts a positive influence on the outcome of IVF but 218 also because it has a positively affects a couple’s quality of life"is far-fetched and unsupported by the results of this work. No one here has studied the effect of psychological support on treatment outcomes!

minor - lack of spaces before quotations.

Author Response

Comment 1: line 42: The in vitro fertilization is not the final step of infertility treatment it is the first step in many cases.

Answer 1: We agree and we have made the necessary modifications to the manuscript (page 1, paragraph 3, lines 43-35).

Comment 2: I would compare the changes in depression among people who feel guilty for the cause of infertility - men with a male factor with men without a male factor. And healthy women with women with a strong infertility factor.

Answer 2: We respectfuly disagree. The study was not designed to address these issues so we cannot offer any insight in this respect at this time.

Comment 3: Also, the difference in years of effort should significantly affect the rate of depression. Therefore, the study group is heterogeneous and should be divided and studied separately.

Answer 3: We agree to an extent. We had addressed this issue indirectly since we had constructed a multivariable logistic regression model in which BDI>16 before IVF onset proved to be a significant and independent predictor of IVF success eventehough the number of previous IVF cycles was also included in the model. We had made changes to the manuscript in order to clarify that this is a multivariable model (pages 2 and 3, lines 95 through 98, page 5, paragraph 2, line 149 and the caption of Table 2).

Comment 4: The conclusion that  "one must consider psychological support paramount for couples undergoing IVF, not just because it exerts a positive influence on the outcome of IVF but 218 also because it has a positively affects a couple’s quality of life"is far-fetched and unsupported by the results of this work. No one here has studied the effect of psychological support on treatment outcomes!

Answer 4: We agree that the wording is to strong. However there might be benefit form psychological support in these patients. Hence we have changed the wording to reflect that. We completely agree that this stuy did not deal with this question so this is more a future research direction statement than a conslusion. See page 6, paragraph 5, lines 223 through 226.

Comment 5: minor - lack of spaces before quotations

Answer 5: We agree and have made all the necessary corrections.